# Development of a Self-Care Scale for Compound Caregivers

**DOI:** 10.3390/healthcare12232438

**Published:** 2024-12-04

**Authors:** Yuka Iwata, Maya Minamizaki, Yuka Kanoya

**Affiliations:** 1Department of Community Health Nursing, Graduate School of Medicine, Yokohama City University, Yokohama 236-0004, Japan; 2Department of Gerontological Nursing, Graduate School of Medicine, Yokohama City University, Yokohama 236-0004, Japan

**Keywords:** scale development, caregiver, self-care, multiple caregiving roles

## Abstract

**Background**: Japan and other nations are experiencing rising demands for care, owing to declining birth rates and aging populations. This particularly affects compound caregivers, people who provide informal care for multiple individuals. Compound caregivers face unique challenges and higher risks of physical and mental health problems. However, existing self-care scales do not cover their needs. This cross-sectional study aimed to develop and validate the Self-Care Scale for Compound Caregivers (SCSCC) to assess specific self-care practices and promote targeted support. **Methods**: A cross-sectional study was conducted through an online survey of 400 compound caregivers. Reliability was assessed via item analysis, exploratory and confirmatory factor analyses, and Cronbach’s alpha. Criterion validity was evaluated using a self-rated health scale. **Results**: Overall, 400 compound caregivers participated in the study. After item analysis, we excluded nine items, leaving eight for factor analysis. Exploratory factor analysis revealed a single-factor structure explaining 51.5% of the variance. Confirmatory factor analysis showed a good model fit after modifications (goodness of fit index = 0.964, adjusted goodness of fit index = 0.932, comparative fit index = 0.980, root mean square error of approximation = 0.054). The final version of the SCSCC demonstrated strong internal consistency (Cronbach’s alpha = 0.859) and was positively correlated with self-rated health (r = 0.387, *p* < 0.001). **Conclusions**: The SCSCC is a reliable tool for assessing self-care in compound caregivers, supporting health promotion, interventions, and policies to improve their health within the community care system.

## 1. Introduction

The demographic shift toward an aging population, combined with declining birth rates, has significantly affected family structures and caregiving dynamics. In Japan and other countries, the traditional extended family model has been undergoing significant changes due to urbanization and the trend toward smaller family units. The number of family members available to share caregiving responsibilities has, therefore, decreased. This has made it more likely that individuals will be required to provide informal care for two or more people with health issues in the home, a concept known as being a compound caregiver [1].

In this study, we defined compound caregivers as those who provide care for at least one individual requiring daily assistance due to health problems, as well as another person—including a healthy child or another individual requiring care—regardless of whether they live with the care recipient. Approximately 40% of caregivers are compound caregivers, and their numbers are expected to rise [2,3].

General caregivers provide daily assistance, medical support such as administering medications, and emotional support to care recipients [4]. Caregiving is physically and mentally demanding, placing a burden on many caregivers. George and Gwyther (1986) defined caregiver burden as “the physical, psychological, emotional, social, and financial problems experienced by family caregivers” [5]. As the community-based integrated care system develops, the prevalence of caregiver burden is expected to increase among informal caregivers, such as family members and relatives. Amid the growth in the number of caregivers, one particularly pressing issue in recent years is the presence of health issues in compound caregivers. Compound caregivers are more likely to experience physical, mental, economic, and social health issues compared with caregivers who care for a single individual [6]. For example, previous studies have reported a higher incidence of metabolic syndrome [7], increased risk of stroke [8], and a higher likelihood of cardiovascular diseases [9] among compound caregivers. Compound caregivers are also more prone to anxiety, overload, fatigue, and high levels of psychological stress [10,11]. This makes self-care for preventing or managing the onset and progression of diseases and symptoms particularly critical for family caregivers [12]. However, compound caregivers often encounter substantial challenges that hinder their ability to practice self-care. Time constraints are a particular concern because attending to the needs of multiple care recipients leaves limited opportunities for personal health maintenance or rest. It is also widely recognized that family caregivers generally prioritize the health needs of care recipients over their own self-care [13]. Financial burdens, including medical expenses and the potential loss of income due to reduced working hours, further restrict their access to self-care resources.

The World Health Organization (WHO) defines self-care as “the ability of individuals, families, and communities to promote health, prevent disease, maintain health, and cope with illness and disability, with or without the support of a healthcare provider” (WHO, 2022). Existing theories on self-care describe it as a complex and dynamic process involving lifelong activities aimed at maintaining health, recognizing symptoms when they arise, and managing those symptoms [14]. Riegel and colleagues categorized self-care into three components: health promotion and treatment adherence (self-care maintenance), body listening and symptom recognition (self-care monitoring), and taking action to manage signs and symptoms (self-care management) [14]. Self-care is closely linked to an individual’s awareness of their health and their willingness to engage in behaviors that promote quality of life [15,16]. Its enhancement has been recognized as an international priority.

Various self-care scales have been developed for diverse populations, including the general adult population [17,18], individuals with hypertension [19], heart failure [20], type II diabetes [21,22], children [23], and family caregivers [12]. Existing self-care frameworks and scales for family caregivers provide valuable tools for assessing and promoting self-care practices. For example, the Self-Care of Informal Caregivers Inventory is based on the Middle-Range Theory of Self-Care of Chronic Illness, providing a robust theoretical framework [12]. These tools contribute to understanding general self-care but may not fully address the unique challenges and needs faced by compound caregivers. No scales have been validated for use with compound caregivers, and no specific scale currently exists to evaluate the distinct self-care practices of this group. Engaging in self-care practices is essential for compound caregivers to maintain both their physical and mental health. Regular self-care can help alleviate stress, reduce the risk of chronic illnesses, and enhance emotional well-being. By prioritizing their own health, caregivers can strengthen their resilience and caregiving capacity, ultimately leading to improved outcomes for both them and the care recipients. Addressing the self-care needs of compound caregivers is therefore critical, not only for their individual well-being but also for ensuring the sustainability of caregiving within the broader community. A self-care scale that is both reliable and valid for compound caregivers should be developed.

The objective of this study was to develop and validate a self-care scale specifically for compound caregivers, the Self-Care Scale for Compound Caregivers (SCSCC) (see Appendix A). By conducting both development and validation phases, we sought to ensure the scale’s reliability and applicability within this specific population. The idea was that this scale could be used to evaluate the unique self-care behaviors of compound caregivers in Japan. Given the high risk of disease and symptom development among this group, targeted self-care support is essential. By quantifying the self-care practices unique to compound caregivers, this scale will be A valuable tool for evaluating self-care promotion programs and designing tailored interventions for compound caregivers.

## 2. Materials and Methods

### 2.1. Study Design

This was a cross-sectional study to develop a scale and test its reliability and validity. We adopted a two-phase process to systematically and rigorously develop the scale. Phase 1 involved creating items to evaluate the self-care behaviors of compound caregivers based on a literature review and expert opinions. This step was crucial to ensure the content validity of the scale and to accurately reflect the concepts to be measured. Phase 2 consisted of conducting a survey among compound caregivers to verify the reliability and construct validity of the developed scale. By structuring the scale development in these two phases, we were able to confirm the theoretical and practical effectiveness of the scale, establishing a foundation for a reliable assessment tool.

### 2.2. Phase 1: Developing the Instrument

The first step in developing the SCSCC was a critical review of the literature. Articles were identified from PubMed and Ichushi-Web using the theme “self-care” among family caregivers who care for multiple individuals. The following search terms were used: “family”, “caregiver”, and “self-care” and the MeSH terms “caregiver, family”, and “self-care”. These searches yielded nine articles [10,11,15,16,17,18,24,25,26,27]. The inclusion of an article was based on two criteria: (1) the article was related to the research experiences of family caregivers who care for multiple individuals requiring care, and (2) the article was associated with existing self-care scales. These articles, and the researchers’ experiences, were used to construct a first draft of the SCSCC containing 29 items.

To ensure the validity of the content, we conducted an expert review process involving six experts. The experts were selected from among peer support staff and family caregivers (two key informants and four compound caregivers identified through an association for this group). The experts were invited to pre-test the initial draft of the scale and evaluate each item using two approaches. First, they rated the relevance of each item for self-care among compound caregivers using a four-point Likert-type scale with responses scored as follows: 1 = very important, 2 = somewhat important, 3 = somewhat unimportant, and 4 = not important at all. An additional option, “Unknown”, was also included as a response in case it was difficult to understand any items. Items were retained if they were rated as “very important” or “somewhat important” by at least 80% of the experts. Items that received mixed ratings or comments regarding clarity were reviewed and revised as necessary. Items rated as “somewhat unimportant” or “not important at all” by more than 50% of the experts were considered for removal. Second, the experts were asked to respond to each item as though they were participants to assess whether the items were answerable. Any item that at least one expert was unable to answer was identified for potential removal. This provided a second draft of the SCSCC that included 18 items.

### 2.3. Phase 2: Validating the Instrument

#### 2.3.1. Participants and Settings

A survey was conducted among 14,000 compound caregivers who were registered as monitors with the Internet survey company NEO Marketing, Inc., Tokyo, Japan. NEO Marketing is a trusted and certified company that has obtained the Privacy Mark.

The questionnaire was distributed to compound caregivers who met the following criteria: (1) they were responsible for the care of more than one person, (2) they were aged 20–79 years old, and (3) they consented to cooperate in this study. The participants were screened to identify compound caregivers using the question: “Are you a compound caregiver, someone who is responsible for the care of more than one person? It does not matter whether the care recipients live with you”. Participants who met these criteria were invited to complete the survey.

Data were collected between 22 August and 27 August 2024. Sampling and data collection aimed to obtain at least 360 usable samples, and the questionnaire survey was closed after 400 responses had been collected.

#### 2.3.2. Measures

The participants’ demographic characteristics included age, sex, living arrangements, relationship to the individuals requiring care, number of individuals requiring care, length of time since becoming a compound caregiver, whether caregiving services are used, and employment status.

Participants were then asked to complete the modified 18-item version of the SCSCC. The SCSCC was scored on a four-point Likert-type scale of 0 = not at all, 1 = not much, 2 = sometimes, and 3 = very often.

To assess the concurrent validity of the scale, the participants were also asked to complete a numerical rating scale (NRS) for self-rated health. The choice of an NRS as a method of self-reporting is grounded in its distinctive capabilities. An NRS is a quick, inexpensive, and convenient self-rated outcome that can capture the person’s overall perception of their health [28]. The NRS for self-rated health requires individual participants to subjectively define their health from 0 (not at all) to 10 (enough) on a ladder device. High scores indicate a high level of self-rated health. In this study, we expected that higher SCSCC scores would correlate with higher scores for self-rated health.

#### 2.3.3. Statistical Analysis

IBM SPSS ver. 28.0 and Amos 29.0 (IBM Corp., Armonk, NY, USA) were used to perform all statistical analyses. An item analysis and exploratory factor analysis were conducted to evaluate the reliability and convergent validity of the SCSCC. An item analysis was performed to assess the quality of each item and to determine whether any items should be excluded from the scale, using the following statistical criteria and cutoff parameters. First, we examined the distribution of responses. Items where one response option was selected by more than 85% of participants were considered to have limited variability and were candidates for exclusion. Limited variability indicates that the item may not effectively discriminate between different levels of self-care behaviors among participants. Second, skewness and kurtosis were assessed. Items with absolute values of skewness or kurtosis greater than 1.0 were considered to deviate significantly from a normal distribution. Third, we calculated the item–total correlations. Items with a corrected item–total correlation coefficient of less than 0.60 were considered for removal. A low item–total correlation suggests that the item does not correlate well with the overall scale and may not be measuring the same underlying construct. Fourth, a good–poor analysis (discrimination index) was conducted. Items that did not show significant differences between the highest-scoring group (top 25% of total scores) and the lowest-scoring group (bottom 25% of total scores) were considered to lack discriminatory power. If an item cannot distinguish between high and low scores, it may not contribute meaningfully to the scale’s ability to measure variations in self-care behaviors. Lastly, item–item correlations were examined. Items with very high correlations (above 0.50) with other items were considered for redundancy and possible removal.

After the item analysis, the total sample was randomly divided into two samples for cross-validation. The total sample of 400 was randomly divided into two split samples (groups 1 and 2) for cross-validation [29]. An exploratory factor analysis (EFA) was performed on group 1, and a confirmatory factor analysis (CFA) on group 2.

An EFA with principal factor analysis with non-rotation was performed on the development sample (Group 1) to assess the underlying factor structure of the SCSCC. Factors with eigenvalues greater than 1.0 were retained, and the scree plot was examined to confirm the number of factors. Items with factor loadings ≥ 0.50 were considered significant. A CFA was then conducted to verify the factor structure identified in the EFA. The following model fit indices were used to assess the adequacy of the model. The goodness of fit index (GFI) measures how well the hypothesized model fits the observed data, with values greater than or equal to 0.90 indicating a good fit between the model and the data. The adjusted goodness of fit index (AGFI) adjusts the GFI for model complexity, and values greater than or equal to 0.90 are considered acceptable. The comparative fit index (CFI) compares the fit of the hypothesized model to a baseline model; values greater than or equal to 0.90 suggest an acceptable fit, while values of 0.95 or higher indicate an excellent fit. The root mean square error of approximation (RMSEA) estimates the discrepancy between the model and the population covariance matrix per degree of freedom. Values less than or equal to 0.08 indicate a reasonable fit, and values less than or equal to 0.060 indicate a good fit. These indices were chosen because they collectively provide a comprehensive assessment of model fit, evaluating absolute fit (GFI, AGFI), incremental fit (CFI), and parsimonious fit (RMSEA), as recommended by psychometric literature [30,31]. Criterion-related validity was examined using the self-rated health score. Internal consistency reliability was evaluated by calculating Cronbach’s alpha coefficient for the SCSCC, with alpha ≥ 0.70 considered acceptable.

### 2.4. Ethical Approval

This research was conducted in accordance with the 1964 Declaration of Helsinki (and its amendments) and the ethical guidelines for life sciences and medical research involving human subjects presented by the Ministry of Health, Labour and Welfare of Japan. The Institutional Review Board of the Medical Department of Yokohama City University School approved this study on 1 May 2023 (No. F230400006).

## 3. Results

### 3.1. Demographic Characteristics

In total, 400 individuals returned the questionnaire, and all 400 (100.0%) met the criteria for inclusion. Table 1 shows the respondents’ demographic characteristics. Compound caregivers ranged in age from 21 to 79 years old, with an average age of 53.7 years old (standard deviation, SD = 12.9). Overall, 69.0% were male, 76.3% were employed, and 79.8% used care services. Each compound caregiver was caring for an average of 2.2 people (SD = 0.5; range 2 to 5). In total, 72.5% of care recipients were the compound caregiver’s parents, 30.5% were parents-in-law, 15.0% were spouses, 14.5% were siblings, and 20.8% were children. The average length of time since becoming a compound caregiver was 4.3 years (SD = 5.9).

### 3.2. Item Analysis

Table 2 shows the item analysis results. Items 1, 2, 4, 5, 7, and 11 had corrected item–total correlation coefficients of less than 0.50, indicating poor correlation with the overall scale. These items were excluded because they did not contribute significantly to the internal consistency of the scale. The correlation coefficient between items 9 and 15, 11 and 12, 13 and 14, 14 and 15, 15 and 16, 15 and 18, 16 and 18, and 17 and 18 were all higher than 0.6. Items 13, 15, and 18 exhibited high inter-item correlations with other items, suggesting redundancy. These items were excluded to eliminate redundancy and improve the scale’s discriminative ability. After excluding these nine items (items 1, 2, 4, 5, 7, 11, 13, 15, and 18), the nine remaining items (items 3, 6, 8, 9, 10, 12, 14, 16, and 17) were subjected to an exploratory factor analysis (principal factor analysis with non-rotation.)

### 3.3. Factor Structure

Table 3 shows the factor loading for the exploratory factor analysis. The latent root criteria (eigenvalues > 1.0) and the scree plot indicated either a one- or two-factor model because of the way the slope leveled off twice. Upon initial exploratory factor analysis of nine items, Item 3 displayed a factor loading in the 0.40 range. It was, therefore, excluded, and the analysis was rerun with the remaining eight items. The first factor explained most of the variability in the eight items, which suggested that the SCSCC had good conceptual consistency (50.6% of the variance explained). Figure 1 shows the factor loading for the confirmatory factor analysis of the SCSCC. The initial model had GFI = 0.938, AGFI = 0.889, CFI = 0.942, and RMSEA = 0.089, which did not represent a good data–model fit. The model fit was improved after modifying with the modification indices and adding error correlations for items 10 and 12 (GFI = 0.964, AGFI = 0.932, CFI = 0.980, and RMSEA = 0.054), which satisfied the appropriate criteria in all subjects (Figure 1). The final version of the SCSCC, therefore, had eight items with a single-factor structure.

### 3.4. Internal Consistency and Validity

The Cronbach’s alpha coefficient was 0.859 for the final version of the SCSCC, showing that the scale had sufficient internal consistency. The final version of the SCSCC was also positively correlated with self-rated health (r = 0.387; *p* < 0.001).

## 4. Discussion

The novel contribution of this research was the development of a new self-care scale for use among compound caregivers to help them look after their health. The SCSCC had adequate reliability and validity. The confirmatory factor analysis model verified the factor validity (GFI = 0.964, AGFI = 0.932, CFI = 0.980, RMSEA = 0.054) and factor correctness of a set of eight observed variables within one factor. The Cronbach’s alpha coefficient for the final version of the SCSCC was 0.859, indicating high internal consistency reliability. Cronbach’s alpha values range from 0 to 1, with higher values suggesting that the items in a scale are measuring the same underlying construct consistently. A value above 0.80 is generally considered indicative of good reliability in social science research [32]. Therefore, the Cronbach’s alpha of 0.859 found in this study shows that the eight items of the SCSCC are closely related and reliably assess the self-care behaviors of compound caregivers. The criterion-related validity was 0.387 between the SCSCC and the NRS for self-rated health. The SCSCC was therefore judged to be a sufficient, reliable, and valid scale that is capable of effectively assessing self-care for compound caregivers.

Compound caregiving is more likely to have a greater impact on the caregiver’s health than providing care for a single care recipient [6]. This highlights the necessity of self-care practices to promote well-being and prevent stress and health issues. This study identified eight specific domains of self-care. These included Item 16, “I delegate tasks instead of taking on everything myself”, Item 10, “I know my limits”, and Item 17, “I utilize care and daily life support services”. All of these reflect self-maintenance strategies to mitigate caregiving burdens. Previous studies have shown that many programs for family caregivers provided by healthcare professionals primarily focus on psychological and educational support, as well as stress reduction [33]. In contrast, this study suggests that directly reducing caregiver burden and distributing caregiving tasks, among others, may be essential components of effective health support for family caregivers. Items such as Item 9, “I connect with friends or peer supporters who can recognize when I am not feeling well”, and Item 12, “I recognize the stress I have accumulated”, represent self-monitoring practices that facilitate early detection of stress. Previous studies have highlighted the importance of noticing physical and emotional changes as a key element of family caregivers’ self-care, which aligns with our findings [12]. Items such as Item 8, “I acknowledge my effort as a caregiver”, Item 14, “I have access to advice on maintaining my health”, and Item 6, “I make sure that I take time to refresh myself regularly”, all reflect strategies for managing stress and exhaustion. Positive reappraisal, a key stress-coping mechanism, can help family caregivers manage negative feelings [34,35]. Positive caregiver resources, such as optimism and resilience, are associated with lower levels of burden and improved quality of life [36,37]. These resources can potentially be enhanced through interventions involving human interaction [35]. In summary, previous studies have identified that general self-care has three key aspects: health promotion and treatment adherence (self-care maintenance), body listening and symptom recognition (self-care monitoring), and taking action to manage signs and symptoms (self-care management) [14]. The items in the SCSCC (Self-Care Scale for Compound Caregivers) were found to comprehensively address these aspects. Compared with existing self-care scales, such as the Self-Care Inventory-Revised (SCI-R) [25] and the Self-Care of Informal Caregivers [12], the SCSCC is uniquely tailored to address the specific challenges faced by compound caregivers. The SCI-R focuses on self-care behaviors in individuals managing chronic illnesses, and the CSCS assesses general self-care practices among caregivers. Neither scale fully captures the complexities associated with caring for multiple individuals. This makes the SCSCC a valuable addition to the existing tools, expanding the capacity of researchers and practitioners to assess and support compound caregivers effectively.

The development of the SCSCC has significant practical implications for healthcare professionals and policymakers involved in supporting compound caregivers. By providing a reliable and valid tool specifically designed for this population, the SCSCC enables the early identification of caregivers who are at risk of neglecting their self-care because of the demands of caring for multiple individuals. This identification is crucial for planning targeted interventions aimed at reducing stress and improving caregiver well-being. By systematically incorporating the SCSCC into caregiver support programs, organizations can monitor the effectiveness of interventions over time. The scale allows for the evaluation of changes in self-care behaviors, enabling adjustments to be made to interventions to better meet caregivers’ needs. Ultimately, the use of the SCSCC may lead to improved health outcomes for caregivers and ensure the sustainability of caregiving within communities.

This study had a few limitations. First, the study design was cross-sectional, and the study, therefore, could not clarify any association between the SCSCC and clinically assessed health outcomes among compound caregivers. A prospective study is needed to determine the scale’s predictive validity. Second, the study’s sample was limited to compound caregivers in Japan, which may affect the generalizability of the findings to other cultural contexts. Cultural differences can influence caregiving roles, perceptions of self-care, and access to support services. It is, therefore, essential to validate the SCSCC in different cultural settings. Third, this study did not investigate the differences between this and other self-care scales. Further studies are needed to examine the discriminant validity of the SCSCC. Fourth, the study did not examine other psychometric properties, such as convergent validity and known-group validity, which could further enhance the robustness of the scale. Future studies should use additional psychometric methods to thoroughly assess the validity and reliability of this instrument [38,39].

## 5. Conclusions

This study contributes to the assessment and identification of compound caregivers who require help to adapt their caregiving roles and improve their health. The SCSCC is a novel instrument with good psychometric properties for assessing self-care among compound caregivers in Japan. Unlike existing self-care scales, the SCSCC addresses the unique challenges of caring for multiple individuals simultaneously, introducing new items that reflect essential self-maintenance strategies not captured in other scales. By comprehensively encompassing the three key aspects of self-care—maintenance, monitoring, and management—the SCSCC enhances the accuracy and relevance of assessing self-care behaviors among compound caregivers. Its use can help to identify caregivers who may benefit from additional support, ultimately enhancing their well-being and the quality of care provided to recipients. To maximize the scale’s potential impact, it is important to validate and adapt it in and for diverse cultural and demographic contexts. However, it may be useful to promote practices, interventions, and health policies to support compound caregivers within a community care system.

## Figures and Tables

**Figure 1 healthcare-12-02438-f001:**
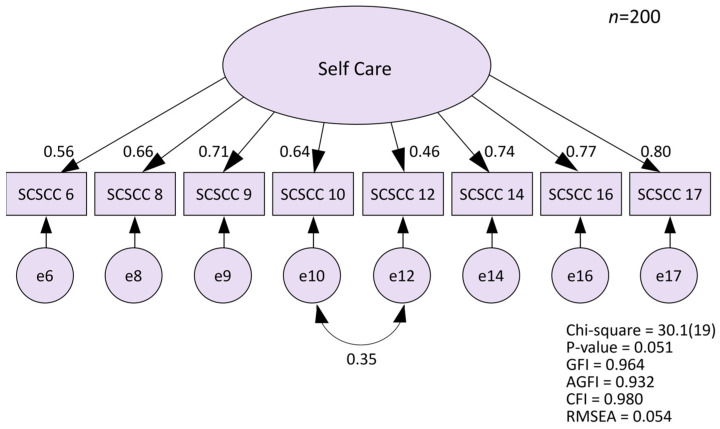
Confirmatory factor analysis of The Self-Care Scale for Compound Caregivers.

**Table 1 healthcare-12-02438-t001:** Demographic characteristics of compound caregivers.

			*N* = 400
		Number or Mean ± SD ^a^	% or Range
Age (years)		53.7 ± 12.9	21–79
	20–29	20	5.0
	30–39	48	12.0
	40–49	71	17.8
	50–59	113	28.2
	60–69	109	27.3
	70–79	39	9.8
Sex	Male	276	69.0
	Female	124	31.0
Living arrangements	Living alone	56	14.0
	With spouse	95	23.8
	With children	14	3.5
	With spouse and children	123	30.8
	With children and grandchildren	74	18.5
	Other	38	9.5
Providing care to	Parent	290	72.5
	Parent-in-law	122	30.5
	Spouse	60	15.0
	Son/daughter	83	20.8
	Sibling	58	14.5
	Other	28	7.0
Number of individuals requiring care		2.2 ± 0.5	2–5
	2	340	85.0
	3	49	12.3
	4	4	1.0
	5	7	1.8
Length of time since becoming a compound caregiver (years)		4.5 ± 5.9	0.4–40
	<1	41	10.3
	1–3	222	55.5
	4–6	71	17.8
	7–9	18	4.5
	10 or more	48	12.0
Use a caregiving service	Yes	319	79.8
	No	81	20.3
Employed	Yes	305	76.3
	No	95	23.8
Self-rated health ^b^		5.9 ± 2.1	0–10

^a^ SD: standard deviation. ^b^ Self-rated health: score from a numerical rating scale for self-rated health.

**Table 2 healthcare-12-02438-t002:** Item analysis of The Self-Care Scale for Compound Caregivers.

									*N* = 400	
No.	Item	Population Distribution ^a^				Kurtosis ^b^	Skewness ^b^	C-I ^c^	G-P ^d^	I-T ^e^
		0	1	2	3					
1	I am aware that my own health is important in looking after my family	7.0	14.2	46.3	32.5	0.000	−0.737	-	0.000 **	0.422 **
2	Sometimes, I prioritize myself over my family	10.8	33.5	42.5	13.3	−0.582	−0.137	-	0.000 **	0.407 **
3	I try to make time for myself whenever possible	5.5	24.5	54.3	15.8	−0.028	−0.392	-	0.000 **	0.511 **
4	I eat regular meals	4.0	24.0	46.3	25.8	−0.469	−0.341	-	0.000 **	0.460 **
5	I get enough sleep	6.3	36.8	41.0	16.0	−0.597	−0.010	-	0.000 **	0.443 **
6	I make sure I take time to refresh myself regularly	4.0	27.8	53.5	14.8	−0.129	−0.252	-	0.000 **	0.591 **
7	I am aware that my actions benefit my family	2.8	19.5	57.5	20.3	0.167	−0.392	-	0.000 **	0.471 **
8	I acknowledge my effort as a caregiver	6.5	31.8	45.3	16.5	−0.489	−0.164	-	0.000 **	0.624 **
9	I connect with friends or peer supporters who can recognize when I am not feeling well	11.3	40.8	36.8	11.3	−0.563	0.063	15	0.000 **	0.573 **
10	I know my limits	3.8	28.2	51.7	16.3	−0.252	−0.209	-	0.000 **	0.589 **
11	I can identify when I am feeling down	5.0	23.5	53.5	18.0	−0.066	−0.390	12	0.000 **	0.497 **
12	I recognize the stress I have accumulated	4.5	19.8	56.5	19.3	0.201	−0.488	11	0.000 **	0.513 **
13	I can adjust my care system to see a physician when needed	7.2	36.8	41.5	14.5	−0.549	−0.031	14	0.000 **	0.623 **
14	I have access to advice on maintaining my health	8.8	34.0	42.0	15.3	−0.583	−0.111	13/15	0.000 **	0.690 **
15	I receive advice from friends and peer supporters on my health	13.8	36.8	34.3	15.3	−0.801	0.030	9/14/16	0.000 **	0.699 **
16	I delegate tasks instead of taking on everything myself	10.0	31.5	46.8	11.8	−0.439	−0.236	15/18	0.000 **	0.701 **
17	I utilize care and daily life support services	12.3	26.3	44.8	16.5	−0.633	−0.294	18	0.000 **	0.571 **
18	I ask reliable individuals for help when necessary	10.0	28.0	49.0	13.0	−0.387	−0.326	15/16/17	0.000 **	0.704 **

**: *p* < 0.001; C-I: correlation of item; G-P: good–poor analysis; I-T: item–total correlation. Exclusion criteria for the item analysis: ^a^: The percentage of at least one of the responses is more than 85% of the sample. ^b^: Absolute value of skewness or kurtosis was less than −1 or greater than 1. ^c^: Correlation was over 0.6. ^d^: Difference in the average score between the highest-scoring group and the lowest-scoring group is not significant (*p* ≥ 0.05). ^e^: The correlation coefficient between the item and the total of all the items (with the exception of the item) is <0.5.

**Table 3 healthcare-12-02438-t003:** Exploratory factor analysis of the Self-Care Scale for Compound Caregivers.

		*n* = 200
No.	Item	Total Scale Communality
16	I delegate tasks instead of taking on everything myself	0.784
14	I have access to advice on maintaining my health	0.724
8	I acknowledge my effort as a caregiver	0.680
9	I connect with friends or peer supporters who can recognize when I am not feeling well	0.679
10	I know my limits	0.675
17	I utilize care and daily life support services	0.607
12	I recognize the stress I have accumulated	0.563
6	I make sure that I take time to refresh myself regularly	0.545
Eigenvalue		4.045
Cronbach’s alpha		0.859

Principal factor analysis with non-rotation.

## Data Availability

The data that support the findings of this study are available from Yokohama City University, but restrictions apply to the availability of these data under the Japan Personal Information Protection Law. The data were used under license for this study and are not publicly available. Data are, however, available from the first/corresponding authors upon reasonable request and with permission of Yokohama City University.

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
