# Peer review of "Development of a Self-Care Scale for Compound Caregivers"

_healthcare, 2024, doi:10.3390/healthcare12232438_

Round 1

Reviewer 1 Report

Comments and Suggestions for Authors

  1. It is unclear what a compound caregiver is and why this scale is necessary. The authors should provide more information in the introduction.
  2. Are there any other inclusion and exclusion criteria for the sample?
  3. The authors should clarify whether they used the same sample for both the Exploratory Factor Analysis (EFA) and Confirmatory Factor Analysis (CFA).
  4. Were any other types of validity tested? For example, convergent validity and known-group validity.
  5. Is there any justification for connecting items 10 and 12 in the CFA?
  6. The authors should mention that more psychometric methods will be used in the future to test the psychometric properties of this instrument. Consider citing: 10.3389/fpsyg.2020.00810; 10.3109/09286586.2010.545501

Author Response

Please find attached a response letter, which includes our point-by-point responses to each of your comments.
Your comments were highly insightful and enabled us to greatly improve the quality of our manuscript.

Reviewer 2 Report

Comments and Suggestions for Authors

Abstract

The abstract effectively presents the study's purpose, highlighting the need to create a self-care scale tailored to compound caregivers and adequately summarizing the main findings. However, there are several ways to enhance clarity and precision.

To enrich the abstract, it would be beneficial to include specific quantitative data, such as the Cronbach’s alpha value, the results from factor analysis (both exploratory and confirmatory), and the correlation with self-rated health. This would allow readers to quickly grasp the key findings. Additionally, consider condensing the methodology description, focusing only on the most critical phases of the scale's development and validation. Finally, a more direct and precise language, omitting minor details, would increase the summary’s impact and make it easier to understand.

 Introduction

The introduction provides a solid context, addressing population aging and the consequent increase in caregiving demand, with special attention to the unique challenges faced by compound caregivers. This section clearly justifies the need for a specific evaluation tool, supported by relevant references that underscore the importance of the issue.

This subsection effectively highlights the study's importance and justifies the development of a scale tailored to compound caregivers. However, it would be helpful to incorporate recent studies to reinforce the current relevance of the problem within the field of self-care. Additionally, a brief discussion of existing self-care frameworks or scales for caregivers is recommended. This addition would situate the scale within the context of available tools, helping to clarify how this proposal complements and expands upon existing literature.

The study’s objective is clearly stated, though it could benefit from a more explicit phrasing that mentions both the development and validation phases of the scale. This adjustment would give readers a complete view of the study’s intent from the outset.

Methods

The methods section is well-structured, providing a clear outline of each phase of the scale’s development and validation, including participant selection, item development, and analysis techniques, which enhances the study's replicability.

The cross-sectional design and scale construction phases are appropriately described. However, adding a brief justification for choosing this design and how it meets the scale’s validation needs would be beneficial.

The scale development process is explained step-by-step, from literature review to item drafting and pretesting with experts. It would be beneficial to elaborate on the expert feedback process, detailing the specific criteria used to adjust or remove items, which would add transparency to the process.

This subsection (instrument validation) includes details on participant sampling criteria and the measurement tools used. It would be an improvement to expand on the reasons for excluding certain items after item analysis, clarifying cutoff parameters for each statistical criterion. This would strengthen methodological justification and transparency in decision-making.

The use of exploratory and confirmatory factor analysis is suitable for evaluating the scale's structure. However, it would be helpful to delve deeper into the criteria used for model fit (e.g., GFI, AGFI, CFI, RMSEA values) and explain why these specific indices were chosen. Additionally, a brief mention of additional tests for internal reliability (e.g., item-total correlations) would contribute to a more complete understanding of the scale’s statistical robustness.

Results

The results section provides a thorough account of findings, including participant demographics, item analysis results, and the final factor structure of the scale.

The sample description is comprehensive, but it could benefit from a visual summary, such as a table or age distribution chart, to aid interpretation and highlight key participant characteristics.

The exclusion of items is clearly detailed, though additional explanation for each exclusion (e.g., low correlation coefficients or factor loadings) would enhance analytical rigor. A table presenting factor loadings for each item in the exploratory factor analysis would allow readers to better assess item consistency.

The internal consistency results, reported using Cronbach’s alpha, are appropriate, and the validity analysis offers valuable insights into the scale’s utility. To improve comprehension, it is suggested to provide a brief interpretation of the statistical results (e.g., explaining the practical implications of a Cronbach’s alpha of 0.859) and how these findings contribute to the scale’s robustness.

Discussion

The discussion emphasizes the scale’s contribution to improving self-care assessment for compound caregivers, underscoring its applicability in guiding specific support interventions.

The interpretation of findings is clear and well-supported by prior studies. However, the discussion could be expanded to cover the practical implications of the results, describing how the scale could be used in planning interventions to reduce stress and improve caregiver well-being.

A more detailed analysis of how the new scale compares to other self-care tools is recommended, as this would highlight the novelty and relevance of the scale in the field of compound caregiver assessment.

Although some limitations are mentioned, it would be beneficial to provide a more in-depth discussion and propose solutions or strategies for addressing them in future studies. For example, the limitation of the cross-sectional design could be mitigated by longitudinal studies, and the scale's validity could be evaluated in other cultural contexts to examine its applicability beyond Japan.

A brief section dedicated to the scale's practical implications for health professionals would enrich the study’s applied value, highlighting its utility as a tool for assessing self-care and guiding individualized support strategies for compound caregivers in various contexts.

The conclusion effectively summarizes the study’s accomplishments, emphasizing the scale’s utility in identifying compound caregivers who may need additional support. To strengthen the conclusion, it would be beneficial to include a final remark on the importance of validating and adapting the scale in diverse cultural and demographic contexts, which would open new avenues for research and reinforce the tool’s potential for international impact.

Author Response

(The authors gave the same response as above.)

Round 2

Reviewer 1 Report

Comments and Suggestions for Authors

The authors have addressed all my concerns. The only comment is I suggest authors include the references reviewer suggested in round one review, as they provide good examples of psychometric analysis using multiple methods within a single study.

Author Response

To the comments of Reviewer 1

 The authors have addressed all my concerns. The only comment is I suggest authors include the references reviewer suggested in round one review, as they provide good examples of psychometric analysis using multiple methods within a single study.

 Response: We appreciate the reviewer’s valuable feedback and suggestions. As recommended, we have included the references suggested during the first-round review. These references provide excellent examples of psychometric analysis using multiple methods within a single study, which we believe enriches the context and supports the methodology of our manuscript.

Line. 365-363

“Future studies should use additional psychometric methods to thoroughly assess the va-lidity and reliability of this instrument [38,39].”

[38] Massof, R.W. Understanding Rasch and Item Response Theory Models: Applications to the Estimation and Validation of Interval Latent Trait Measures from Responses to Rating Scale Questionnaires. Ophthalmic Epidemiol. 2011, 18, 1–19, doi:10.3109/09286586.2010.545501.

[39] Xu, R.H.; Wong, E.L. yi; Lu, S.Y. jun; Zhou, L.M.; Chang, J.H.; Wang, D. Validation of the Toronto Empathy Questionnaire (TEQ) Among Medical Students in China: Analyses Using Three Psychometric Methods. Front. Psychol. 2020, 11, 1–11, doi:10.3389/fpsyg.2020.00810.

Reviewer 2 Report

Comments and Suggestions for Authors

The quality of the manuscript has improved significantly. I suggest publishing it in its current version.

Author Response

We deeply appreciate Reviewer 2’s feedback on our revised manuscript. It is truly gratifying to know that the quality of our work has significantly improved to meet your expectations. Your insightful comments and thoughtful suggestions throughout the review process have been instrumental in refining the manuscript, and we are immensely grateful for your support. Thank you for your recommendation to publish the manuscript in its current version.